# “Knitting Together the Lines Broken Apart”: Recovery Process to Integration among Japanese Survivors of Intimate Partner Violence

**DOI:** 10.3390/ijerph191912504

**Published:** 2022-09-30

**Authors:** Sachiko Kita, Kiyoko Kamibeppu, Denise Saint Arnault

**Affiliations:** 1Department of Family Nursing, Division of Health Sciences & Nursing, Graduate School of Medicine, The University of Tokyo, Tokyo 1130003, Japan; 2Department of Health Quality and Outcome Research, Global Nursing Research Center, Graduate School of Medicine, The University of Tokyo, Tokyo 1130033, Japan; 3Graduate Programs in Family Nursing, International University of Health and Welfare, Tokyo 1078402, Japan; 4Department of Health Behavior and Biological Sciences, University of Michigan School of Nursing, Ann Arbor, MI 48109, USA

**Keywords:** gender-based violence, intimate partner violence, trauma recovery, trauma integration, positive psychology, psychological health

## Abstract

Objective: This study used narrative interviewing and grounded theory analysis to discover the phases of trauma integration for Japanese women who had experienced intimate partner violence (IPV). Method: We interviewed 23 Japanese women who had experienced IPV using the Clinical Ethnographic Narrative Interviews (CENI) from November 2017 to September 2018 in Tokyo, Japan. The data from 11 participants who had achieved trauma integration using the Trauma Recovery Rubric were analyzed using a modified grounded theory approach. Results: Six phases of the trauma integration journey after IPV were discovered: (1) Chaos, (2) Burning out, (3) Focusing, (4) Challenging, (5) Deepening insights, and (6) Re-building. The survivors described the processes of exploration of themselves and their reactions to their concurrent challenges. They focused on finding ways to protect, re-discover, and re-embrace themselves by healing from physical, psychological, and spiritual distresses. They also rebuilt relationships with others and acquired knowledge and skills to achieve a new life. Notably, the primary components of their recovery processes were changes in self-perception and self-interpretation of the trauma itself and its impacts on one’s life and within oneself from multiple angles. In addition, traditional Japanese norms and gender roles, such as beliefs concerning the “ideal life of a woman” and fear of not behaving differently from others, profoundly influenced their recovery process. Conclusion: This study highlights the importance of incorporating individuals’ cultures and their phase, needs, and personal self-development timeframes when developing trauma integration interventions.

## 1. Introduction

Intimate partner violence (IPV) is a severe, highly prevalent international health issue. IPV is described as behaviors that exert power and control over an intimate partner using physical, psychological, sexual, and other violence types [1,2]. Notably, one in three women has experienced IPV worldwide [3]. In Japan, 31.3% of women have experienced some form of violence from a partner [4], and 5.0% for severe violence threatening their life [5]. IPV has long-term impacts on the bio-psycho-social health of the victims. These include Posttraumatic Stress Disorder (PTSD), mood and anxiety disorders, self-harm, eating and sleeping disorders, and unexplained physical symptoms (e.g., body pain and vaginal bleeding) [6,7,8,9]. One study of Japanese survivors of IPV [10] reported that 76% experienced moderate-severe symptoms of PTSD, 68.4% had anxiety, and 50% had depression for an average of seven years after the IPV incidents. Although these findings necessitate paying more attention to IPV survivors, effective interventions that meet their needs and enhance their recovery are rare [2]. This scarcity of trauma recovery data is possibly due to limitations of previous studies, such as their reliance on only cross-sectional survey data. A holistic narrative investigation is necessary to understand the overall picture of their health and needs after IPV. This study aims to examine the recovery processes of Japanese IPV survivors, hearing their needs and the methods they use to integrate their trauma and move into healthier lives.

There has been an increase in scholarly attention to the need to promote positive change and enhance strength among survivors of violence. Previous studies have reported that such survivors heal by engaging in several interrelated activities. These healing activities include rebuilding self-love, reconnecting with others and the world, and increasing their sense of appreciation, hope, fulfillment, and peace. Survivors also report that processing and responding to painful feelings (i.e., fear, anxiety, loneliness, guilt, and worthlessness) helps them heal [11,12,13,14,15]. Such positive changes in self-perception, relationships with others, and general philosophy of life as a result of surviving a highly traumatic event could be explained by “trauma integration,” which involves the concepts of trauma healing or posttraumatic growth [11,14,16]. Walsh [15] described three key processes after traumatic experiences: (1) deriving meaning from the traumatic loss experience; (2) positive outlook: (3) hopes, future dreams, and transcendence. Previously, Herman [17] suggested three stages, including (1) safety and stabilization, (2) remembrance and mourning, and (3) re-connection and reintegration. However, these were general concepts regarding recovery after many kinds of trauma, such as domestic violence, community violence, war-related trauma, and major disasters. In addition, Walsh [15] focused on the entire family rather than traumatized individuals. Thus, the trauma recovery process among survivors of IPV may differ from those of the previous studies and understanding the unique recovery process after IPV may advance our understanding and treatment. 

The Trauma Recovery Rubric (TRR) was developed to identify and classify seven types of trauma recovery pathways using data from survivors of gender-based violence (including IPV) in four countries: America (*n* = 24), Japan (*n* = 24), Turkey (*n* = 13), and Greece (*n* = 14) [18]. These pathways of trauma recovery are conceptualized as a non-linear continuum, which includes (1) normalizing, (2) minimizing, (3) consumed/trapped, (4) shutdown or frozen, (5) surviving, (6) seeking and fighting for integration, and (7) finding integration/equanimity. Seeking integration and finding equanimity share the characteristics of regaining self-mastery and health, such that the survivor has a degree of self and trauma understanding and mastery. This rubric defines integration as achieving self-understanding and achieving a sense of peace and security within themselves. The seeker has hope that a state of peace is possible but is still experimenting and practicing. The integrated person is settled and emotionally secure because they know that recovery is a life’s work. They feel equanimity or mental calmness, composure, and an even temper, especially in difficult situations [18]. Although these international collaboration studies [18] suggested that the recovery trajectories included these seven types, we understand that each is unique and requires additional detail to understand each recovery pathway’s characteristics. In these Japanese data, because of the recruitment method (being recruited by institutions that serve survivors of IPV), we have a sample size of integrated women that allows us to examine the integration pathway in more detail. This present study examines the qualitative reports from survivors of IPV who had achieved such integration states and aims to illuminate the processes they went through on their journey toward this state as a first step in identifying a more complete picture of one of these diverse recovery processes. Here, we seek to discover *how* IPV survivors experienced the internal changes on their road to integration and the internal changes they made along the way. Identifying and facilitating such changes is critical for holistically understanding the IPV recovery journey and can help us suggest more detailed, timely, and effective strategies that meet survivors’ needs at all stages.

Meanings and strategies for recovery are strongly influenced by survivors’ sociocultural backgrounds, such as cultural and religious beliefs on gender roles, support systems, and interpretation of violence in society [19,20]. Japanese survivors of IPV (like any survivor) experience a culturally unique recovery process for trauma integration. Sinko et al. [19] reported the differences among the vital components of trauma integration between the two countries, finding that Irish women strongly valued their mothering responsibilities. In contrast, American women valued personal growth and tenacity. Notably, in Japan, survivors of IPV may experience difficulties when seeking help, gaining sufficient formal and informal support, becoming independent, and accepting the traumatic experience, possibly due to cultural norms, gender roles, and traditional beliefs. Such beliefs include familial solidarity, discrimination against victims and single mothers, and Japanese principles of virtue [21,22,23]. Nearly half of the IPV victims who later recounted their stories did not seek help or disclose their experience of trauma to anyone, and only 38.3% reported sexual assault [4]. IPV survivors with children (i.e., single mothers) may also experience severe socio-economic difficulties. One study found that the annual single-mother household income equals about approximately 23,000 in US dollars [24]. Another study reported that 68.6% of survivors do not work full-time [10]. Taken together, we can see that Japanese IPV survivors may experience unique recovery processes due to immense cultural and economic challenges. How they achieve trauma integration can catalyze the development of effective interventions that incorporate their sociocultural backgrounds. Moreover, using a narrative interview method and grounded theory analysis, this study can enhance our understanding of how culture and sociocultural context relate to recovery processes. 

## 2. Methods

### 2.1. Study Design, Period, and Setting

The design of this study was a qualitative descriptive study using a narrative interview method conducted from November 2017 to September 2018 in Tokyo, Japan.

### 2.2. Study Participants

Women who had experienced IPV in the past were potential participants in this study. Eligible participants were recruited according to the following inclusion criteria: (a) 18 to 60 years old; (b) had sufficient Japanese ability to conduct the interview. Women also had to self-identify as having been victimized by IPV or any other violence type and be separated from the abuser within the past year. Survivors who were suffering from severe mental illness were excluded from this study. According to the inclusion and exclusion criteria for this research, the directors of two IPV service centers, a non-profit support institute for single mothers and an attorney’s office that provided legal services for victims of IPV in Tokyo, as the collaborators of this study, selected and recruited eligible women from their client base. They knew their situations, backgrounds, and mental status well. Twenty-eight eligible women were recruited, and twenty-six participants agreed to this study. Of those, 23 participants completed the interview, while 3 women did not come to the interview. Using the TRR [18], eleven participants out of the twenty-three participants were classified as “integration/equanimity (integrated).” This study reports only data from those eleven women, aiming to characterize the detailed processes that led them to integration. 

### 2.3. Procedure

Eligible women were recruited by the directors of the four institutes and received a brief explanation about this study using a flyer. If they were interested in the study, they were asked to contact the researcher (SK) via email. Once they emailed the researcher, they received details about this study, such as the procedure, interview method, and ethical considerations using the consent form. If they agreed to participate, they were asked to sign the consent form and return it to the researcher. After they agreed to participate, they were asked their available date for the face-to-face interview. Before the interview, the study was explained to them using the consent form once again by the researcher(s), and confirmed their willingness to agree to participate in this study. The interviews were conducted in a private room at the university or the shelters and were recorded using two digital voice recorders. The interviews averaged about 2 h (ranging from 1.5 to 3.5 h). After the interview, information regarding accessible IPV and psychological support resources and incentives (approximately 30 in US dollars) was provided. One week after the interview, they were phoned by the researcher to assess if they experienced any physical and emotional problems or changes. If they wanted help, they were referred to appropriate social resources, such as psychological health services. 

### 2.4. Interview

The Clinical Ethnographic Narrative Interview (CENI) was used. The CENI was developed to gather culturally relevant illness experience, meaning, and help-seeking and then adapted for use with trauma survivors [25]. The CENI is a 90 min semi-structured interview composed of four activities: *Social network mapping*, *body map*, *lifeline,* and *card sort*. The *social network* mapping helps the participant describe all help-seeking behaviors within the social context. The *body map* elicits physical and psychological experiences [26]. The *lifeline* facilitates a retrospective overview of distress and identifies linkages between experiences and actions [27,28,29]. The *card sort* references a low point in the lifeline and facilitates the description and organization of physical and psychological feelings [30,31]. Finally, the participant describes causal interpretations, social significance, and help-seeking actions, assisting them in identifying their beliefs, meanings, patterns, and processes [25,32]. 

### 2.5. Qualitative Analysis

Only data from eleven women classified as “integrated” using the TRR were analyzed to identify an especially detailed recovery process to integration. This study used a modified grounded theory analysis approach [33,34]. We extracted data describing the recovery process to trauma integration. We compared the data with the previously extracted data to confirm the similarities and differences between the data to produce concepts. Next, we considered the relationships between the produced concepts and the produced subcategories, which consisted of similar concepts. Finally, we produced categories after considering the similarities and differences between these subcategories and organized them chronologically. Experts supervised the entire process of the analyses in qualitative research (DSTA and KK), and the Japanese categories, subcategories, and concepts were translated into English by two research assistants and were supervised by the co-authors. The names of the categories, subcategories, and concepts were determined throughout careful and repeatable readings of the transcripts and discussions among the authors. In addition, the anonymized transcripts after removing all personal information during the interview were used for the analyses. 

### 2.6. Ethical Considerations

This study protocol was approved by the Ethical Committee of the Graduate School of Medicine at the University of Tokyo (No. 11756-2). 

## 3. Results

### 3.1. Characteristics of the Participants

Almost all the participants (*n* = 11) were over 41 years old (*n* = 10) and working (*n* = 8). Most of them had graduated from a technical school or junior college (*n* = 7). All of them had experienced IPV from ex-husbands, and two had also experienced childhood abuse and/or sexual assault. Of the eleven survivors, eight had been diagnosed with PTSD (*n* = 5), mood disorder (*n* = 3), dissociative disorder (*n* = 1) and/or asthma (*n* = 1; see Table 1). 

### 3.2. Journey to Integration

Six IPV recovery process categories were extracted: (1) Chaos, (2) Burning out, (3) Focusing, (4) Challenging, (5) Deepening insights, and (6) Re-building. The women described those six categories as stages or phases they passed through on their journey to their current recovery state (see Figure 1). The x-axis was time, and the y-axis was emotional capacity. The time from the first to the last phase varied widely between the participants, ranging from 1.5 years to over 20 years. Therefore, the length of any given phase and the ratios of time of each phase were not the same; for instance, one survivor described that she stayed in the phase of Burning out for one year, and another said this phase lasted for over five years. These variations seemed to depend on numerous recovery conditions, such as their health status, the complexity of trauma and circumstance, and social support.

Additionally, the time frames of each phase were not necessarily demarcated. Phases frequently overlapped with others, moving back and forth. However, all the participants finally described the integration experience and the phases mentioned above. These phases represent the progression from initial escape toward their current state of trauma integration. In the following, we present the phases’ descriptions and highlight the subthemes that characterized these phases using italics. 

***Phase 1: Chaos.*** Chaos was the first phase women experienced as soon as they escaped from or were separated from their abusive partners. During the phase, they described feeling that they were in a chaotic situation, even though they seemed to be functioning. They also described feeling unsafe and disconnected from their body, mind, and social contexts. This phase was a highly tense period where they dealt with several problems following separation from their abuser, such as moving to a new place to live, managing their daily lives alone, and initiating legal divorce proceedings. The women reported that day-to-day troubles and feelings dominated their lives, leaving them no energy and time to pay attention to their internal feeling and problems. One woman said: “I didn’t know what to do, but I had to decide on an apartment and a school for my children quickly…I still don’t remember how I could deal with these things” (ID J0010). 

In this phase, women experienced *deep sorrow for losing their familiar environment, friends, and work*. One woman reflected, “After leaving, I lost my house, friends, and job simultaneously. It was the most painful experience in my life” (ID J0028). They also described *disappointment with responses from persons from whom they sought help*. They were disappointed when they faced discrimination and rejection from parents, friends, and public officers when they sought help. Another woman said: “My parents told me, ‘…you embraced us because you got divorced and needed a place’ when I escaped to their house. I felt I was a troublemaker” (ID J0001). Women also described *fears and panic of being chased by abusers* and *intense anxiety for the children, who were also affected by violence and separation.* For example, one woman discussed their fears by saying: “A couple of days later, after escaping, my husband called me. At the time, I had no feeling except a strong fear of my husband, so I panicked” (ID J0005). 

On the other hand, the women described a solid will to move forward from this chaotic situation. Additionally, ten women experienced gratitude for support from persons they sought help from and were surprised by positive changes in children soon after the separation. One woman said, “Before I left the house, I had a lot of feelings of anxiety, fear, and hesitation…but after leaving, I strongly felt I can’t come back anymore, and I have to move forward” (ID J0013). Another woman reflected, “I was so surprised that my child could sleep soundly the whole night since the day we left” (ID J0017). 

***Phase 2: Burning out.*** Women experienced the phase of Burning out after several of their day-to-day living and survival needs were met, and they started to live in a safe environment. This phase was referred to as the period that they faced and struggled with the severe and numerous impacts of violence, such as their physical, psychological, and social symptoms and emotional pain. Moreover, they suffered from feeling they had become different from others or had changed significantly, referring to it as “…falling from the bridge where everyone crosses” (ID J0013). They expressed that this phase was the hardest in their life after IPV. Nevertheless, they recognized this phase was a significant and necessary step toward becoming aware of and reconsidering their habitual ways of living and thinking.

Many women experienced unexplained physical symptoms in this phase, such as stomach pain, headache, nausea, palpitation, excessive weight loss, severe coldness, itchiness, or a long-lasting cough. One woman mentioned, “I was wondering what this palpitation is…and was very worried thinking that ‘I might die soon.’ I took medicine, but it didn’t work” (ID J0004). They also reported *a lack of confidence and high pressure to proceed onward.* For example, one woman noted, “My husband used to tell me, ‘you are useless and can’t do anything’ for a long time, so I didn’t think I could decide my future by myself” (ID J0013). They experienced *loneliness and isolation for facing their feelings and experiences alone*. Consequentially, they did not disclose or seek help from anybody. *Four women reported explosive anger toward the abusers who had changed and ruined their lives*. For example, one woman stated, “I repeatedly thought why he had done such terrible things to me…I wanted to pay (him) back. I was always irritated, and it made me so exhausted” (ID J0018).

Additionally, eight women experienced *bewilderment to their child’s emotional explosion,* such as aggressive behaviors, withdrawal, anger, shame, resistance to changing lifestyles, and regression due to the high stress of leaving a familiar environment and their father. One woman said, “My adolescent son started to be violent towards me because he didn’t like the situation of using a welfare service to live” (ID J0004). Another woman mentioned, “My (4-year-old) daughter … always chased me around and became insecure for 24 h. I think she exploded her suppressed trauma” (ID J0027).

Although they experienced these intense feelings, survivors also reported feeling suddenly powerless and sluggish. One woman described it as if “…an inflated balloon was popped” (ID J0009). They felt *guilty and impatient for being lethargic*. They felt too lonely and exhausted to think about their problems and future. They forgot their values, blamed themselves for being unable to live their lives properly, and felt impatient with themselves for their inability to work despite wanting to. Survivors summarized these feelings, saying: “I felt I was lazy and useless for myself being unable to do anything” (ID J0017); “I didn’t have power left to think about myself” (ID J0009); and “I felt very lonely compared to my children who seemed to adapt to this situation smoothly” (ID J0027). 

***Phase 3:******Focusing.*** This phase occurred when the survivors focused on internal and external causes of their distress experienced in the Burning out phase, such as their habitual ways of thinking and becoming aware of their strong desire to stay true to themselves and to change their lives. In this phase, they described living *“someone else’s life”* as one of the leading causes of their distress, low self-confidence, and sense of inferiority. They also noticed the persistence of gender roles in their environment, as well as in their self-expectations. For example, one woman said: “I noticed I was exhausted because I had clung to the ideal characteristics of a ‘woman,’ ‘wife,’ and ‘mother,’ such that a wife has to make three meals every day and a mother has to smile and be cheerful all the time” (ID J0027). Seven women noticed that they always cared about others’ perceptions, resulting in physical and psychological distress, such as shame, regret, and fear. For instance, another woman mentioned, “When I looked back, I always thought that ‘I have to be a good woman’ from the perspective of my parents, my husband, and the world in general. I felt like I was living according to other people’s wishes” (ID J0018). 

Survivors experienced an *awareness that suppressing physical symptoms and psychological emotions is abnormal.* To survive in a violent and unsafe environment, they tried not to feel any physical symptoms and distressing emotions, such as body pain, anger, unpleasantness, sadness, and loneliness. However, in this phase, they noticed this was abnormal and unhealthy. One woman described, “I had held tons of black stones (*Kuroi-ishi*) in myself, and I had tried to ignore all of them to keep a ‘normal life’ with my husband. However, I noticed that holding the black stones is not healthy for me and (that they) have to be released to the outside” (ID J0010).

Once the survivors became aware of these internal causes of distress, they felt *an awakening desire to stay true to themselves*. They desired to regain their senses and to do so, and they attempted to change their circumstances. One woman said: “I became so calm after I noticed I have freedom (nobody controls me), and I naturally started to think about what I want to do, and what makes me happier and calmer” (ID J0009). 

***Phase 4:******Challenging.*** After becoming aware of the causes of their burning out and discovering their desire to change, survivors moved to the Challenging phase. The Challenging phase was described as the period in which they explored and tried to protect, re-discover, and regain sensibility, living life by healing physical and psychological distress, reconnecting with trusted others, and acquiring knowledge and skill to cope with trauma. However, survivors also simultaneously experienced new challenges, resulting in further distress. In this phase, all the women experimented with new actions to heal themselves and expand their possibilities, such as connecting with or joining a peer support group for IPV, finding or starting a job, and receiving several formal and informal (e.g., traditional) therapies. While forming connections with trusted others, such as peers, a new partner, and family, they experienced *regaining a sense of security and peace with others* and *increasing confidence in their decision-making capacity.* One woman noted, “I feel that peers are very important for me. We don’t deny our feelings and opinions with each other. I feel so easy and secure in the relationships” (ID J0025). Another mentioned, “My ex-husband used to repeatedly say, ‘You are not useful in the society,’ so I had to quit my job. In contrast, I noticed, ‘Oh, I’m very useful in society!’ when I came back to the job” (ID J0013).

Moreover, the *connections they noticed between trauma, body, and mind* were described after acquiring knowledge and skills through books, classes, and peer-support groups, yielding a greater understanding of their experience and symptoms. For example, one woman said, “I didn’t know why, but sometimes I felt pain and discomfort around my uterus, and when I felt it, my tears suddenly came out. But after I recognized that I was raped, I understood that the pain in my uterus came from my trauma and emotions” (ID J0025). 

While embracing the positive changes, survivors also confronted new types of internal conflicts and distress during this phase, and their emotions were “shaken.” For example, they described *helplessness for impulsive and intense anger and fear* concerning their ex-husbands and other people (e.g., their parents and co-workers) who were indifferent towards, ignored, prejudiced, and discriminated against their trauma. One woman said: “I was used to thinking and living like ‘it is all my fault.’ But after I learned and recognized, ‘My trauma is not my fault,’ intense anger and hatred against the abuser and other people around me who didn’t help me suddenly came up. I couldn’t handle these painful feelings.” (ID J0025). In addition, women felt *hopeless and vulnerable to unfairness, prejudice, and discrimination surrounding them* by incessantly facing prejudice and victimization. One survivor mentioned, “When I started to work in a female counseling office for victims of violence, I saw my senior workers who labeled victims as ‘weak and troublesome women’ saying, ‘Victims have some faults (that cause them to be)…abused.’ I felt very sad, vulnerable, and hopeless” (ID J0005). Two survivors also described *facing a core source of distress* they had ignored, such as painful childhood experiences that deepened their understanding of their suffering. One woman expressed it as “…opening a locked box left alone for a long time” (ID J0017). They felt anger, helplessness, regret, chaos, and self-blame when they remembered their experience during childhood. Another woman mentioned, “I noticed my senses of shame, inferiority, isolation, and loneliness came from my childhood experience. My mother used to frequently slap and pinch me when I didn’t do what she wanted me to do. My friends said it was child abuse, but defining the experience is (still) too painful for me because I loved my mother” (ID J0018). 

***Phase 5: Deepening insights.*** After skills, knowledge, and thoughts were accumulated during the previous phase, survivors experienced the phase of Deepening insights. This phase was described as the period when they deepened their insights about themselves and their trauma, deciphered the meaning of their trauma and its impact on their lives, and interpreted why it happened. Such profound insights and philosophies developed gradually among some women but came up suddenly for others. By deepening their insights, survivors discovered connections between their trauma and emotions, their past and current experiences, and their body, mind, and social contexts. One woman described this phase as “…knitting together the lines that had broken apart” (ID J0025). 

First, women began to view their lives, including trauma episodes, from multiple angles, noticing that *life can’t be explained by trauma alone.* Women not only focused on their misery but also remembered and appreciated the meaning and impact of pleasant times in their lives, affirming that trauma was only one part of their lives. One woman stated, “I don’t want to recognize myself as a victim because this is only one part of my life. I have experienced many happy times in my life, such as spending time with my children and doing my work” (ID J0004). Another survivor said, “I recognized that the saddest and the happiest moments were represented by one event in my life, such as the marriage with my ex-husband. So, I can’t judge my life using only one standard” (ID J0017). Additionally, the survivors expanded their perspectives toward people who had hurt them in the past, such as their abusive ex-husband and parents. They *discovered different aspects of the person who hurt them by* viewing these people from more distant and different angles. They recognized that the people who hurt them were not merely “bad people.” One woman said, “Looking back, I find that my dominant mother just didn’t know how to do it. I now know she raised me sternly” (ID J0009). Another said, “After I heard my father used to be abused by his mother, I feel that my father who abused me was also suffering” (ID J0017). 

Survivors *recognized that their trauma had been healed by trusted others*, such as peers, friends, children, new partners, and co-workers. Throughout rebuilding secure and stable relationships with others, survivors felt they had been protected, healed, encouraged, and helped to confront and overcome their challenging period after GBV. One woman said: “My children, my new partner, and I discussed anything freely and equally. Those secure relationships healed my trauma” (ID J0013). Based on their newfound trust in and acceptance by others, women *regained the confidence to stay true to themselves*. Survivors examined their selfhood, discovering their strengths, weaknesses, and habitual thought and action patterns while trying to understand their aggressors. One woman mentioned: “It is better to think about what I want to be and not what other people expect of me. I want to focus on what kind of human being I am and how I want to live” (ID J0027). Six survivors described trying to interpret or discover why GBV happened to them. One woman stated, “I think my loneliness since childhood attracted my ex-husband (abuser) who had had similar experiences” (ID J0027). Another woman said, “I used to be controlled by my mother. Now I feel the controlling relationship just moved from my mother to my ex-husband.” (ID J0009).

When survivors reviewed and interpreted their entire life, all the survivors described *reconnecting with their experiences, body, minds, and social contexts*. They were convinced that their physical and psychological distresses, such as persistent headaches, breathlessness, or a loss of desire to live, were associated with trauma. One woman expressed this by saying, “…my body and mind remember the trauma” (ID J0013). For example, one woman mentioned, “I feel my emotions and memories relating to the past trauma that can’t be released and handled sometimes appear as various physical symptoms” (ID J0013). Another said: “…vague difficulties, such as persistent loneliness and a lack of self-confidence, have been lasting since my childhood. These feelings were related to my experience being with my parents” (ID J0009).

Furthermore, women recognized that their bodies and minds were always interconnected and that they influenced each other. One woman said, “…when I have a headache, I interpret I’m emotionally exhausted, so I try to rest or avoid the stressors” (ID J0005). Another said, “…my shoulder and back pains came from the pressure and stress of work” (ID J0028). While pondering on their entire lives and health, the survivors regained and increased the manageability and predictability of their physical and psychological distresses. They also acquired skills to cope with their distress, such as providing self-compassion with words such as “I can’t get frustrated, so don’t worry. You are doing best” (ID J0018), taking purposeful breaks, sharing feelings with trusted others, or seeking helping professionals (e.g., healer and acupuncture therapist). 

***Phase 6:******Re-building.*** All the women experienced the final phase of Re-building. This phase was described as the period wherein women felt they had integrated the trauma into their lives, embraced their selfhood, and gained confidence in their ability to deal with the trauma impact. During this phase, they could see an overall picture of their selfhood, believe in their strength, and look toward the future. All the women described peace of mind and appreciating life as, “My mind is now calm like the sea” (ID J0009) and “I’m very glad to be alive” (ID J0027).

First, women *believed that trauma is also an important aspect of their lives*. They appreciated meeting or reconnecting with trusted others. Survivors now viewed the trauma as one event in the course of life and recognized that their current life was re-constructed by trauma. One woman mentioned, “It is my life, even if it gets painful” (ID J0010). Another woman stated, “I used to want to erase the hard period of my life, but now I feel that the period is also my life” (ID J0027).

Furthermore, survivors believed that *“the trauma strengthened me.”* Survivors felt they learned essential things from their experiences and became more mature. They understood the feelings of people with similar experiences, devoid of prejudice and discrimination, and they felt they became more compassionate than before the trauma. One woman said: “I have learned a lot from the struggling period. I became a person who understands people’s feelings” (ID J0017). 

Furthermore, survivors *strongly believed in their competence and decisions*. They were proud of themselves that they could overcome such a challenging period and rebuild their lives. One woman stated: “When looking back to my past, I finally overcame so many challenging things in my life. I am proud of myself and believe I can overcome anything the future holds” (ID J0010). Moreover, they *firmly believed that their decisions were not all ‘wrong.*’ These decisions included not only the divorce from their ex-husband but also the smaller decisions in daily life. Women reported that although they did sometimes make mistakes in life, they could always learn something from these mistakes. One woman mentioned, “I made various mistakes and sometimes regretted it, but I think my key decisions, including separating from my husband, were not wrong” (ID J0001). 

After regaining confidence in their competence and decisions, women finally *embraced the**ir lives and the fact that they had experienced trauma*. They decided to live with the trauma, respect their feelings and hopes, avoid being manipulated by others, and recognize that “…it was fine to be me even if I am not perfect” (ID J0027). Survivors affirmed and embraced their way of living and were rarely influenced or shaken by other people anymore. One woman said: “I don’t mind how other people perceive me. I like myself now, even if it may seem weird to other people” (ID J0028). Another woman, who had experienced sexual violence, reported decreased shame and began to enjoy being a woman. Consequentially, these survivors started to *look at the future and not the past,* focusing on enriching their lives more. One woman said: “I don’t know what will happen in the future. However, I always want to look forward to making my life better” (ID J0009).

## 4. Discussion

This study explored the detailed phases of the IPV recovery process among Japanese survivors, discovering the importance of self-interpretation and self-perception. This evidence suggests that one essential trauma integration skill, rebuilding relationships with others, was achieved through a deepened understanding of the trauma itself and its impacts on one’s life and within oneself from multiple angles. Although previous studies indicated more general ideas and elements regarding phases of the recovery process after many kinds of trauma, including domestic violence, community violence, war, and the loss of a family member [15,16,17], this study identified gender and culturally specific recovery processes, as well as the chronological recovery processes leading to integration among Japanese survivors of IPV. These holistic views help us understand *how* survivors of IPV may experience each phase in detail. 

During the earlier stages, such as the phases of Chaos and Burning out, survivors struggled with several physical and psychological distress related to difficulties in interpreting and accepting the trauma. This phenomenon may be strongly influenced by sociocultural pressure and norms related to gender roles; beliefs concerning the “ideal life of a woman” such as supporting their husband and children; fear of not behaving differently from others; and the prejudice faced by victims of violence and single mothers after divorce in the Japanese society [22,23,35]. However, we found that survivors could succeed in re-discovery, rebuilding, and reembracing themselves, even though they remained in Japanese society. Culturally relevant interventions can include encouraging discussing distress and difficulties while respecting their Japanese cultural beliefs. Decreasing self-stigma can be achieved by recognizing that “being different from others” is not shameful. Providers can help survivors affirm their decision to escape (and become single mothers) by acknowledging that protecting their children from violence is paramount. These changes in self-perception can enhance self-acceptance without challenging all Japanese cultural norms.

Furthermore, health professionals and care providers should pay more attention to *how* they should intervene depending on the vulnerability of the phases, especially the Chaos, Burning out, and Challenging phases. For example, in the Chaos phase, participants felt extremely tense when facing and dealing with urgent problems, such as finding a place to live, hiding from abusers, and commencing legal proceedings. Thus, they did not have the energy or time to focus on their symptoms and emotions. Prolonging these insecure and unstable social circumstances after IPV may lead to delaying internal healing and recovery [2,10]. Thus, multidisciplinary interventions, including lawyers, public officers, social workers, and welfare workers, are imperative to address the survivors’ needs and enhance their attention on personal health. During the Burning out phase, survivors experienced several physical symptoms and psychological emotions, which severely impaired their functioning in daily life. Thus, referring them to a health care service and providing intensive health interventions, such as psychotherapy, counseling, medication, and traditional therapy, may be appropriate [2]. Lastly, in the Challenging phase, their emotions are strongly shaken and unstable as the survivor’s endeavor to change themselves and their circumstances while experiencing positive changes, such as regaining confidence in their decision-making capabilities and starting re-connections between trauma and symptoms. Previous studies [14,36,37] indicated that reconnecting with others and gaining supportive social reactions helps bolster feelings of security, validation, and engagement in the change process after IPV. Thus, professionals should focus on social support resources with trusted others. Currently, public health interventions and policies to enhance holistic recovery among survivors of IPV have not yet been implemented in Japan due to limited social and financial resources. A long-term psychological intervention beyond medication is needed, such as peer support and counseling to enhance their holistic recovery while respecting their cultural norms and gender roles, and policies to provide social and financial support for survivors to encourage their utilization of such interventions on a long-term basis.

### Limitations

There were several limitations of this study. First, only the data for Japanese samples who achieved trauma integration were used. The previous study [11] indicated that the recovery process among abused women is strongly influenced by culture. Thus, a future study comparing these with the results of studies in different countries will be necessary to understand cultural similarities and differences in integration processes.

Second, this study used only data from eleven women categorized as “integration/equanimity.” While qualitative analysis is not intended to be generalizable, additional samples are needed to discover if these findings are transferrable to other types of trauma survivors. Thus, an additional study with larger sample sizes and diverse trauma survivors is needed for this study’s generalizability and to discover other survivors’ unique processes.

Third, the features of the participants of this study may be biased due to the recruitment method. The participants of this study were recruited by the directors of the IPV services centers; thus, they have been already connected with and supported psychologically, socially, and economically by the IPV service centers. Thus, the participants might be bio-psycho-socially healthier compared to a general population of survivors of IPV in Japan. Thus, the results of this study should be cautiously interpreted.

Further studies with larger sample sizes from diverse backgrounds in multiple countries may be necessary to improve the generalization and international application of the results of this study. In addition, future studies with survivors showing other types of recovery pathways will be essential to grasp the whole picture of diverse recovery processes after IPV.

## 5. Conclusions

This study aimed to describe detailed phases of the recovery process for trauma integration among Japanese survivors of IPV using ethnographic narrative interviews. The six phases reported for trauma integration are (1) Chaos, (2) Burning out, (3) Focusing, (4) Challenging, (5) Deepening insights, and (6) Re-building. Notably, the unique recovery processes of trauma integration among Japanese survivors were self-interpretation and self-perceptions, which may be culturally specific characteristics. We did find that the process of their exploration and challenges helped them to protect, re-discover, and re-build themselves. The results of this study should be valuable and helpful for clinical professionals and care providers of formal and informal service agencies for IPV to understand the detailed processes, especially for trauma integration among such survivors of IPV, and develop effective interventions that meet their needs, timing, and cultural backgrounds to enhance their trauma integration more efficaciously. In addition, the results of this study may be applicable to survivors of IPV not only in Japan but also the other countries; thus, an additional study with larger and more diverse populations in multiple countries may be necessary to improve the generalizability and international application of this study.

## Figures and Tables

**Figure 1 ijerph-19-12504-f001:**
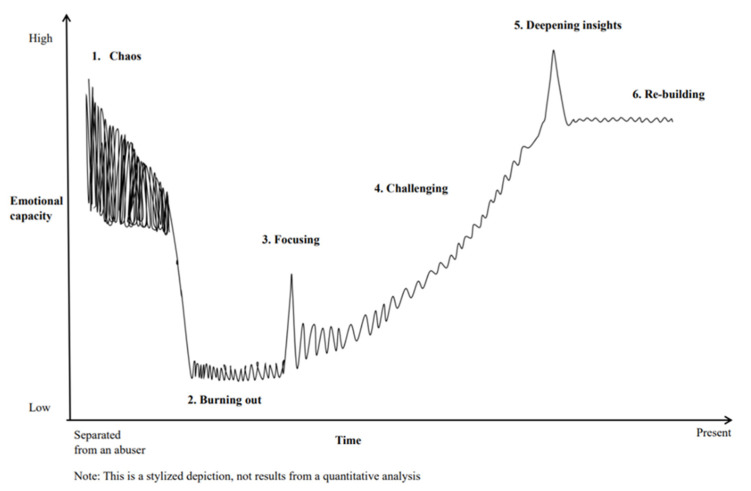
Recovery process to trauma integration.

**Table 1 ijerph-19-12504-t001:** Characteristics of survivors’ who achieved trauma integration.

Demographics	Total (*n* = 11)
Age (years)	
20–30	0
31–40	1
41–50	5
50+	5
Parent	10
Working status	
Working	8
Non-working or public assistance	3
Education	
Technical school or junior college	7
University or higher	4
Trauma history	
Intimate partner violence (IPV) by ex-husbands	11
Complex trauma	2
IPV + Childhood abuse	1
IPV + Childhood + abuse sexual assault	1
History of being diagnosed psychologically and physically	8
PTSD	5
Mood disorder	3
Dissociative disorder	1
Asthma	1

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
