# Peer review of "“Knitting Together the Lines Broken Apart”: Recovery Process to Integration among Japanese Survivors of Intimate Partner Violence"

_ijerph, 2022, doi:10.3390/ijerph191912504_

Round 1
Reviewer 1 Report
This is a report of a study between November 2017 to September 2018 in Tokyo, Japan of women using particular social services who recognized themselves as victims of intimate partner violence who had achieved what the authors refer to as Trauma Integration. There were eleven participants in this narrative research study who were interviewed.
The study is well-documented and the insights of the authors helpful, both in constructing their detailed phases of trauma and regarding the advice they provide to professionals as a result of taking note of these different phases.
For the most part, this report is well-written—although it is clear by the type of mistakes made that two subsections were written by an author or authors not well-acquainted with English.
The authors need to clearly indicate the differences between the phases to trauma recognized by Herman and Walsh and then identify in what way their new phases proposed in this paper enrich or differentiate from the work of Herman and Walsh.
It is evident that this is important research, both because of the paucity of research on this topic and because of the new and helpful insights provided by this study. However, the labels that the authors have decided on for the different phases they have recognized for trauma, in often conflating two very different ideas, are not theoretically helpful. This reviewer has made suggestions for new, one word titles, but the authors may create their own. The point being that the title for the phase should be only one word, even if that word is representing a complex idea.
The final point is that although these authors have created a helpful framework for those working with Japanese women who have suffered IPV (that may, in fact, be internationally applicable) they have done so from the perspective of trauma. This is likely not the perspective of the women who were victims. Their perspective most likely was developing as an individual. As such, since this is narrative research, the focus should be on the perspective of the women, not fitting their perspective into a rubric defined by trauma. If the perspective of the paper is switched from trauma to the “Recovery process to integration” witnessed by these women, it is likely the need for two word labels for the different phases will no longer be necessary.
Some references to methodology used are missing. Furthermore, the citation and reference styles also have to be redone to MDPI style.
Line by line suggested edits
69 Change “Additionally” to “Previously”. Please also indicate how Herman’s definition differs from that of Walsh with respect to the almost twenty year difference in publication dates of their research.
74 Change “. Still, they include” to “that includes”.
74-76 It is unclear how the authors have come up with these divisions in their non-linear continuum and why they differ from those of Herman or Walsh.
86 GBV has not been defined before this use of the acronym.
102 Change “victims did” to “victims who later recounted their stories did”.
105 “mother-child”—Do you mean single mother?
115-116 Please provide a peer-reviewed reference for the type of narrative interview method conducted.
122 How did the authors confirm that the woman had been separated from their abuser within the past year?
123 How did the authors confirm that the survivors did not suffer from severe mental illness?
133 Change “they were explained about this study” to “the study was explained to them”.
134 Change “will” to “willingness”.
163-164 Please describe the reason for using only these women for the analyses.
178 The Ethical Committee of what organization?
182 Change “with” to “to participate in”.
183 If there were 23 participants who completed the interview explain here why the demographic recorded in Table 1 indicates only 11 participants.
183-187 What differentiates the 11 classified as a “survivor” from the 23 classified as a “participant”? That needs to be stated here.
Table 1: Heading should be retitled “Characteristics of Survivors’ Who Achieved Trauma Integration”.
Change “Having a child” to “Parent”.
192-194 How were these categories named? Please include information on how the labels were determined. Confronting does not seem to be the same sort of thing as burning out. One can confront and not burnout. As well, one can burnout without awareness that they are confronting. Furthermore, one can have deepened insights without reconnection and one can reconnect without having any deep insights. Lastly, one can have integration of their thoughts without re-embracing their lives and vice versa.
Figure 1. Please label the x and y axis of the graph. As well, describe in the heading that this is a stylized depiction, not results from a quantitative analysis.
210-211 If they seemed energetic and active, in what way did they feel confused?
215 Why was this phase called “confusion” rather than “struggling”? The actions of the women don’t appear confused.
238 Given that confronting and burning out are quite different and can exist independently and that Figure 1 shows this stage as a bottoming out, perhaps “bottoming out” is a clearer description of this phase.
251 Why is “Participate” used instead of “ID”?
276-279 This description of “awareness” might be better labeled as “focusing”, especially since Figure 1 shows that this is a process, not a single point in time (as awareness seems to indicate).
299 If the authors do believe that this awareness did come at a particular point in time then maybe this phase is better labeled “awakening”.
303-307 The Challenging/Being shaken label is not theoretically helpful as it conflates two things that are very different and can exist on their own. Why was this stage not labeled something like “healing”?
348 Again, it is not theoretically helpful to have a label including two disparate concepts like deepening insights and re-connection when the relationship between them is unclear.
376 Perhaps this stage is better labeled “re-building”.
410 Once again, the double label of two very different terms with a connection that is not obvious is theoretically confusing. Perhaps “re-establishing” is a term that encompasses both.
438-439 It seems that saying the women embraced the trauma is viewing their acceptance of themselves from the perspective of the trauma rather than of the women. It is the women’s acceptance of themselves that is important, not their acceptance of a process that has been categorized as “trauma”.
453 Did the women actually deepen their understanding of the trauma or of themselves? It seems that the important thing was that they faced themselves, not that they came to know about trauma. Trauma was what the researchers were concerned about, not necessarily the women.
455-457 Although this study adds more details, this discussion needs to include information about in what way details were provided to the descriptions of trauma by Herman and Walsh and why this new method of thinking about trauma is more effective.
505 Change “eleven data from women” to “data from eleven women”.
506 Change “It” to “Limiting the participants in this way” and change “might lead to the smaller sample size of this study and be not” to “leads to a smaller sample size and not being”.
507 Change “additionl” to “additional”.
508 Change “samle” to “sample”.
516 A further limitation of this study is that the authors may have recognized the phases to trauma they did as a result of their focus on trauma rather than on the way the women perceived themselves as developing individuals rather than being concerned with trauma per se.
529 The conclusion should also note that this particular phase view of trauma and its implications for professionals might be internationally applicable and further research should be done in this regard.
Author Response
Thank you very much for reviewing our manuscript and providing us your insightful comments and suggestions. Please see the attachment.

Reviewer 2 Report
Thank you for allowing me to review this well-written and conceptualized the study.
The present study explores trauma recovery among IPV victims in Japan. The authors identified several phases leading to recovery. Previous studies on Japanese IPV therapy center on the adaptation of western concepts such as CBT. Thus, the present study explored a novel and interesting topic worthy of dissemination through an academic journal. Nonetheless, I have several recommendations that the authors may consider to improve their manuscript.
Abstract: Brief and concise
Introduction: The introduction was well-written and discusses several recovery frameworks. The authors also numerously mentioned the research gaps and aims of their study.
Methods: clear and concise
Results: Well written, the quotations were well-placed and weaved into the paragraphs. However, Figure 1 is quite confusing: 1. it is not clear what the x and y axis mean; 2. it is not clear what the actual line graph is, it left me guessing whether this is the clarity of thought or cognition or trauma symptom of the study participants. I suggest that legends and axis names be added to this figure.
Discussion: the discussion can be improved by enriching it with the comparison of the similarities and differences of the recovery framework in this study with western models such as Herman's. Additional discussion would allow readers to distinctly understand the uniqueness of your model.
The discussion can also be enriched by including public health programs and policy interventions that can be implemented aside from provider-level care.
There were multiple typographical errors in the limitations section such as line 507 "additionl" and line 515 "generl"
Additionally, the authors can also add research recommendations beyond those that address the present study's limitations.
Conclusion: Well written.
Author Response

(The authors gave the same response as above.)
